# Support Interval for Two-Sample Summary Data-Based Mendelian Randomization

**DOI:** 10.3390/genes14010211

**Published:** 2023-01-13

**Authors:** Kai Wang

**Affiliations:** Department of Biostatistics, University of Iowa, 145 N Riverside Dr., Iowa City, IA 52242, USA; kai-wang@uiowa.edu

**Keywords:** mendelian randomization, summary statistics, SMR, profile likelihood, support

## Abstract

The summary-data-based Mendelian randomization (SMR) method is gaining popularity in estimating the causal effect of an exposure on an outcome. In practice, the instrument SNP is often selected from the genome-wide association study (GWAS) on the exposure but no correction is made for such selection in downstream analysis, leading to a biased estimate of the effect size and invalid inference. We address this issue by using the likelihood derived from the sampling distribution of the estimated SNP effects in the exposure GWAS and the outcome GWAS. This likelihood takes into account how the instrument SNPs are selected. Since the effective sample size is 1, the asymptotic theory does not apply. We use a support for a profile likelihood as an interval estimate of the causal effect. Simulation studies indicate that this support has robust coverage while the confidence interval implied by the SMR method has lower-than-nominal coverage. Furthermore, the variance of the two-stage least squares estimate of the causal effect is shown to be the same as the variance used for SMR for one-sample data when there is no selection.

## 1. Introduction

A main interest in scientific research is to study the causal effect of an exposure *x* on an outcome *y*. When the outcome is continuous, the causal effect is the coefficient *b* in the following regression model:(1)$$y=bx+u+\epsilon_{y},$$
where *u* represents the unobserved factors and $\epsilon_{y}$ is normally distributed with mean 0 and variance $\sigma_{y}^{2}$. Throughout this report, all variables are centered so that the intercept is equal to 0.

When *u* confounds the effect of *x*, the least squares estimate of *b* is biased. Mendelian randomization (MR) is a modern technique for correcting this bias [1,2,3,4,5], thanks to the availability of the large number of genome-wide association studies (GWASs). One appealing feature of the summary-data-based MR methods is that they don’t rely on individual-level data.

MR is an application of instrumental variable (IV) analysis to estimate *b*. IV analysis is able to control for unobserved confounders. MR uses single nucleotide polymorphisms (SNPs) as IVs. Let *g* denote the genotypic score of an SNP. For an IV to be valid, it must satisfy the following assumptions [6]:**Relevance:** It is associated with the exposure *x* (i.e., Cov(g,x)≠0);**Exclusion Restriction:** It affects the outcome *y* only through its association with the exposure; and**Exchangeability:** It is not associated with any confounders of the exposure–outcome association, which implies Cov(g,y)=bCov(g,x).

Under these assumptions, Equation (Equation 1) implies:$$b_{gy}=bb_{gx},$$
where $b_{gy}=Cov\left(g,y\right)/Var\left(g\right)$ and $b_{gx}=Cov\left(g,x\right)/Var\left(g\right)$. Since $b_{gx}$ and $b_{gy}$ can be estimated from the exposure GWAS and the outcome GWAS, respectively, a popular summary-data MR (SMR) estimate of *b* is [2]:$${\hat{b}}_{\mathrm{SMR}}=\frac{{\hat{b}}_{gy}}{{\hat{b}}_{gx}},$$
where ${\hat{b}}_{gy}$ and ${\hat{b}}_{gx}$ are GWAS estimates of $b_{gy}$ and $b_{gx}$, respectively.

In practice, in order to satisfy the relevance assumption on an IV, an SNP is typically selected from the exposure GWAS, often at the genome-wide significance level $p<5\times 10^{-8}$. Hence the selected SNPs are subject to a winner’s curse that leads to a biased effect estimate. This is an issue that has been recognized in the MR literature for some time [7,8,9,10,11,12,13]. A typical solution is to use another GWAS on the exposure to screen for IV SNPs [10,14,15]. However, such a GWAS may not always be available. A simple correction is to transform the false discovery rate to the z-scale [11]. An empirical study on the effect of the winner’s curse on a Mendelian randomization study is presented in [12]. A review of methods to overcome the winner’s curse in the context of genetic association studies is provided in [13].

As a matter of fact, many applications [2,16,17], including those contained in the original paper that proposed SMR [2], do not use another GWAS for screening. The IV SNPs are simply selected from the exposure GWAS and the selection bias is not corrected for in the downstream analysis [2,18,19]. Furthermore, this approach has been generalized to other settings [20,21].

This research is based on the sampling distribution of the estimated SNP effects on the exposure and on the outcome. Despite the large number of subjects used in the exposure GWAS and the outcome GWAS, there are only 4 summary statistics (i.e., two coefficient estimates and their respective standard errors) needed for MR at an IV SNP. The standard asymptotic theory, which requires a large sample size, does not apply since there is only one “observation” at an SNP. To sidestep this issue, we use a support derived from a profile likelihood as an interval estimate for *b* and assess its coverage probability through simulation. Support is the set of parameter values at which the log profile likelihood is a certain unit below the maximum log profile likelihood. It can be considered an extension of the confidence interval. Simulation studies demonstrate that the 2-unit support has robust coverage while the confidence interval implied by the SMR method has lower-than-nominal coverage.

In addition, we point out that the standard error of ${\hat{b}}_{\mathrm{SMR}}$ derived from the delta method is the same as the standard error derived from the standard theory on two-stage least-squares (TSLS) regression for one-sample individual-level data in the absence of SNP selection.

## 2. Materials and Methods

### 2.1. One-Sample Individual-Level Data

In this subsection, we consider one-sample individual-level data where the IV SNP is not selected for its significant *p*-value. In this case, the delta method estimate of $SE({\hat{b}}_{\mathrm{SMR}})$ is the same as the estimate derived from the theory on the TSLS method. The delta method is the method used by SMR [2]. This result indicates another connection between SMR and TSLS, in addition to the connection that ${\hat{b}}_{\mathrm{SMR}}$ is the same as the TSLS estimate of *b*. James E. Pustejovsky proved this relationship in a blog post [22]. Below we provide a similar proof in our current context.

Consider the following GWAS model on exposure *x*:$$x=b_{gx}g+\epsilon_{x},$$
where $\epsilon_{x}$ correlates with the unobserved confounder *u*. The reduced-form equation for *y* is:$$y=b_{gy}g+\left(b\epsilon_{x}+u+\epsilon_{y}\right),$$
where $b_{gy}=bb_{gx}$. Let x, y, and g denote the centered vectors of *n* observations on *x*, *y*, and *g*, respectively. Estimates of $b_{gx}$ and $b_{gy}$ can be obtained as follows: ${\hat{b}}_{gx}=g^{\prime }x/g^{\prime }g$ and ${\hat{b}}_{gy}=g^{\prime }y/g^{\prime }g$. Let $P=gg^{\prime }/g^{\prime }g$. Their second-order moments are estimated by:$$\begin{array}{cc} \hat{Var}\left({\hat{b}}_{gx}\right) & =\frac{n^{-1}x^{\prime }\left(I-P\right)x}{g^{\prime }g},\\ \hat{Var}\left({\hat{b}}_{gy}\right) & =\frac{n^{-1}y^{\prime }\left(I-P\right)y}{g^{\prime }g},\\ \hat{Cov}\left({\hat{b}}_{gx},{\hat{b}}_{gy}\right) & =\frac{n^{-1}x^{\prime }\left(I-P\right)y}{g^{\prime }g}. \end{array}$$
We note that $\hat{Cov}\left({\hat{b}}_{gx},{\hat{b}}_{gy}\right)\neq 0$.

The TSLS estimate of *b* is the least squares estimate of the coefficient in the regression where the response is y and the predictor is Px:$${\hat{b}}_{\mathrm{TSLS}}=\frac{x^{\prime }Py}{x^{\prime }Px}=\frac{x^{\prime }g{\left(g^{\prime }g\right)}^{-1}g^{\prime }y}{x^{\prime }g{\left(g^{\prime }g\right)}^{-1}g^{\prime }x}=\frac{{\left(g^{\prime }g\right)}^{-1}g^{\prime }y}{{\left(g^{\prime }g\right)}^{-1}g^{\prime }x}=\frac{{\hat{b}}_{gy}}{{\hat{b}}_{gx}}={\hat{b}}_{\mathrm{SMR}}.$$
The delta method estimate of the variance of ${\hat{b}}_{\mathrm{SMR}}$ is [2]:$$\begin{array}{cc} V_{\mathrm{delta}} & =\frac{1}{{\hat{b}}_{gx}^{2}}\left[Var\left({\hat{b}}_{gy}\right)+{\hat{b}}_{\mathrm{TSLS}}^{2}Var\left({\hat{b}}_{gx}\right)-2{\hat{b}}_{\mathrm{TSLS}}Cov\left({\hat{b}}_{gx},{\hat{b}}_{gy}\right)\right]. \end{array}$$
Since ${\hat{b}}_{gx}^{2}g^{\prime }g=x^{\prime }Px$, we have:(2)$$\begin{array}{cc} V_{\mathrm{delta}}\; & =\frac{1}{{\hat{b}}_{gx}^{2}}\cdot \frac{1}{ng^{\prime }g}\left[y\left(I-P\right)y+{\hat{b}}_{\mathrm{TSLS}}^{2}x^{\prime }\left(I-P\right)x-2{\hat{b}}_{\mathrm{TSLS}}x^{\prime }\left(I-P\right)y\right]\\ \; & =\frac{n^{-1}{\left(y-{\hat{b}}_{\mathrm{TSLS}}x\right)}^{\prime }\left(y-{\hat{b}}_{\mathrm{TSLS}}x\right)}{x^{\prime }Px}-\frac{n^{-1}{\left(y-{\hat{b}}_{\mathrm{TSLS}}x\right)}^{\prime }P\left(y-{\hat{b}}_{\mathrm{TSLS}}x\right)}{x^{\prime }Px}\\ \; & =\frac{n^{-1}{\left(y-{\hat{b}}_{\mathrm{TSLS}}x\right)}^{\prime }\left(y-{\hat{b}}_{\mathrm{TSLS}}x\right)}{x^{\prime }Px}. \end{array}$$
The last equal sign holds because:$$P\left(y-{\hat{b}}_{\mathrm{TSLS}}x\right)=g{\left(g^{\prime }g\right)}^{-1}g^{\prime }y-\frac{g^{\prime }y}{g^{\prime }x}\cdot g{\left(g^{\prime }g\right)}^{-1}g^{\prime }x=0,$$
where 0 is a vector of 0’s. The right-hand side of Equation (Equation 2) is exactly the estimated variance of ${\hat{b}}_{\mathrm{TSLS}}$ defined in the standard theory on TSLS method [23]. Combining all these results, we have $V_{\mathrm{delta}}=V_{\mathrm{TSLS}}$.

$V_{\mathrm{delta}}$ can not be computed from GWAS summary data as there is no information on $Cov({\hat{b}}_{gx},{\hat{b}}_{gy})$. For the same reason, TSLS can also not be computed from GWAS summary data.

However, when ${\hat{b}}_{gx}$ and ${\hat{b}}_{gy}$ are derived from two independent samples, $Cov({\hat{b}}_{gx},{\hat{b}}_{gy})=0$ and $V_{\mathrm{delta}}$ can be computed from GWAS summary data, as is shown in the SMR method [2]. SMR tests whether the exposure has a causal effect on the outcome using a statistic $T_{\mathrm{SMR}}$ defined by:(3)$$T_{\mathrm{SMR}}=\frac{{\hat{b}}_{\mathrm{SMR}}^{2}}{V_{\mathrm{delta}}},$$
where ${\hat{b}}_{\mathrm{SMR}}={\hat{b}}_{gy}^{2}/{\hat{b}}_{gx}^{2}$ and
$$\begin{array}{cc} V_{\mathrm{delta}} & =\frac{1}{{\hat{b}}_{gx}^{2}}\left[Var\left({\hat{b}}_{gy}\right)+{\hat{b}}_{\mathrm{SMR}}^{2}Var\left({\hat{b}}_{gx}\right)\right]. \end{array}$$
On the other hand, the TSLS method is not defined for two-samples MR although there are some extensions [24].

### 2.2. Two Independent Samples with a Selected SNP

In this subsection, we consider two-sample MR where the IV SNP is selected from the exposure GWAS. The purpose of this selection is to ensure that the IV SNP is associated with the exposure. This practice is commonly used in empirical MR studies [2,16]. The selection criterion for an SNP is typically $\left|{\hat{b}}_{gx}\right|/SE\left({\hat{b}}_{gx}\right)\geq \tau$ for a prespecified τ. For the genome-wide significance level $5\times 10^{-8}$, τ=5.45131.

The summary statistics used in an MR analysis are ${\hat{b}}_{gx},SE\left({\hat{b}}_{gx}\right),{\hat{b}}_{gy}$, and $SE({\hat{b}}_{gy})$. To simplify notations, they will be denoted by *x*, $\sigma_{x}$, *y*, and $\sigma_{y}$, respectively, and $b_{gx}$ and $b_{gy}$ will be denoted by $\mu_{x}$ and $\mu_{y}$, respectively. We ignore the sampling variation in $\sigma_{x}$ and $\sigma_{y}$ as they typically are derived from GWASs of very large sample sizes. A similar assumption is made elsewhere, for instance, [25]. In these notations, *x* and *y* have the following sampling distributions, respectively:$$x∼CN\left(\mu_{x},\sigma_{x}^{2}\right),\;\;y∼N\left(\mu_{y},\sigma_{y}^{2}\right),$$
where CN(·,·) stands for a conditional normal given $|x/\sigma_{x}|\geq \tau$ and N(·,·) a normal distribution. The distribution function CN(·,·) was used to construct an approximate conditional likelihood for estimating $\mu_{x}$ [26]. The term “approximate” comes from the fact that the distributions of *x* (prior to selection) and *y* are approximately normal.

Let $\alpha_{1}=-\tau -\mu_{x}/\sigma_{x}$, $\alpha_{2}=\tau -\mu_{x}/\sigma_{x}$, and
$$\begin{array}{cc} A & =\mathrm{Pr}\left(x/\sigma_{x}\geq \tau \right)+\mathrm{Pr}\left(x/\sigma_{x}\leq -\tau \right)\\ \; & =1-\Phi \left(\alpha_{2}\right)+\Phi \left(\alpha_{1}\right), \end{array}$$
where Φ(·) is the cumulative distribution function of standard normal. The density function of *x* is:$$\frac{1}{A\sigma_{x}}\phi \left(\frac{x-\mu_{x}}{\sigma_{x}}\right),$$
where ϕ(·) is the density function of standard normal. The expected value of *x* is:(4)$$\mu_{x}+\frac{\sigma_{x}}{A}\left[\phi \left(\alpha_{2}\right)-\phi \left(\alpha_{1}\right)\right]$$
and its variance is:$${\tilde{\sigma }}_{x}^{2}=\sigma_{x}^{2}\left\{1+\frac{\alpha_{2}\phi \left(\alpha_{2}\right)-\alpha_{1}\phi \left(\alpha_{1}\right)}{A}-\frac{{\left[\phi \left(\alpha_{2}\right)-\phi \left(\alpha_{1}\right)\right]}^{2}}{A^{2}}\right\}.$$
Note that ${\tilde{\sigma }}_{x}^{2}$ is no longer constant; its value depends on $\mu_{x}$. When there is no selection, τ=0 and $\alpha_{1}=\alpha_{2}$. The mean and variance reduce to $\mu_{x}$ and $\sigma_{x}^{2}$, respectively.

Since *x* is independent of *y*, the likelihood function $L(\mu_{x},\mu_{y})$ is:$$L\left(\mu_{x},\mu_{y}\right)=L_{x}\left(\mu_{x}\right)L_{y}\left(\mu_{y}\right),$$
where $L_{x}\left(\mu_{x}\right)$ and $L_{y}\left(\mu_{y}\right)$ are the likelihood functions based on *x* and *y*, respectively.

The likelihood function $L_{y}\left(\mu_{y}\right)$ is:$$L_{y}\left(\mu_{y}\right)=\frac{1}{\sigma_{y}}\phi \left(\frac{y-\mu_{y}}{\sigma_{y}}\right).$$
The MLE of $\mu_{y}$ is apparently ${\hat{\mu }}_{y}=y$.

The likelihood function $L_{x}\left(\mu_{x}\right)$ is:$$L_{x}\left(\mu_{x}\right)=\frac{1}{A\sigma_{x}}\phi \left(\frac{x-\mu_{x}}{\sigma_{x}}\right).$$
Its score equation is:(5)$$x=\mu_{x}+\frac{\sigma_{x}}{A}\left[\phi \left(\alpha_{2}\right)-\phi \left(\alpha_{1}\right)\right].$$
This equation determines the MLE ${\hat{\mu }}_{x}$ for $\mu_{x}$. However, there is no explicit form for ${\hat{\mu }}_{x}$. The MLE ${\hat{\mu }}_{x}$ can be obtained by maximizing $L_{x}\left(\mu_{x}\right)$ numerically.

From Equations (Equation 4) and (Equation 5), *x* is an unbiased estimate of the mean of the conditional normal distribution for *x*. However it is biased for $\mu_{x}$ as the mean shown in Equation (Equation 4) is a nonlinear function of $\mu_{x}$. Similar comments are made elsewhere [26].

Since $\sigma_{x}>0$ and A>0, Equation (Equation 5) indicates that when x>0$\mu_{x}$ must be positive. Otherwise $x-\mu_{x}$ would be positive and $\phi \left(\alpha_{2}\right)-\phi \left(\alpha_{1}\right)$ is negative (because $\alpha_{1}$ is closer to 0 than $\alpha_{2}$ is). There would be a contradiction. Because $\mu_{x}>0$ implies that the second term of Equation (Equation 5) is positive, there is $x>{\hat{\mu }}_{x}>0$. Following the same logic, when x<0, there is $x<{\hat{\mu }}_{x}<0$. In either case, the naïve Wald ratio y/x underestimates $b=\mu_{y}/\mu_{x}$. The MLE of *b*, denoted by $\hat{b}$, is $\hat{b}=y/{\hat{\mu }}_{x}$. $\hat{b}$ is biased. The expectation of $\hat{b}$ is $E\left(y\right)E\left(1/{\hat{\mu }}_{x}\right)\neq b$ because $E\left(1/{\hat{\mu }}_{x}\right)\neq 1/\mu_{x}$.

Figure 1 shows the ${\hat{\mu }}_{x}/\sigma_{x}$ as a function of $x/\sigma_{x}$. The larger the value of $x/\sigma_{x}$, the smaller the absolute difference $|x/\sigma_{x}-{\hat{\mu }}_{x}/\sigma_{x}|$. When $|x/\sigma_{x}|\geq 7.5$ (corresponding to a *p*-value less than or equal to $6.38\times 10^{-14}$), the absolute difference $|x/\sigma_{x}-{\hat{\mu }}_{x}/\sigma_{x}|$ is <0.05 and seems to be negligible.

Under the null:$$H_{0}:b=\mu_{y}/\mu_{x}=0,\mu_{x}\neq 0,$$
there is $L\left(\mu_{x},\mu_{y}\right)=L_{x}\left(\mu_{x}\right)L_{y}\left(0\right)$. The MLE of $\mu_{x}$ is equal to ${\hat{\mu }}_{x}$ determined by Equation (Equation 5). Under the alternative:$$H_{1}:b=\mu_{y}/\mu_{x}\neq 0,\mu_{x}\neq 0,$$
the MLE of $\mu_{y}$ is *y* and the MLE of $\mu_{x}$ is still ${\hat{\mu }}_{x}$. The likelihood ratio statistic for testing $H_{0}$ against $H_{1}$ is:$$T=2\log\frac{L({\hat{\mu }}_{x},y)}{L({\hat{\mu }}_{x},0)}=2\log\frac{L_{y}\left(y\right)}{L_{y}\left(0\right)}=\frac{y^{2}}{\sigma_{y}^{2}}∼\chi_{1}^{2}.$$
This is the “conditional test” we proposed previously [9]. It is more powerful than the SMR statistic shown in Equation (Equation 3).

The SMR statistic $T_{\mathrm{SMR}}$ shown in Equation (Equation 3) does not taking into account the effect of the selection of the IV SNP on the inference. The variance of *x* is no longer $\sigma_{x}^{2}$ because *x* is selected. Even if $\sigma_{x}^{2}$ is replaced by the variance ${\tilde{\sigma }}_{x}^{2}$ of $CN(\mu_{x},\sigma_{x}^{2})$, the resulting statistic is less powerful than the *T* statistic: Replacing ${\hat{b}}_{\mathrm{SMR}}$ by $y/{\hat{\mu }}_{x}$ and $\sigma_{x}$ by ${\tilde{\sigma }}_{x}$ in Equation (Equation 3), a modification of the SMR statistic would be:$${\tilde{T}}_{\mathrm{SMR}}=\frac{y^{2}/{\hat{\mu }}_{x}^{2}}{\left(\sigma_{y}^{2}+y^{2}/{\hat{\mu }}_{x}^{2}\cdot {\tilde{\sigma }}_{x}^{2}\right)/{\hat{\mu }}_{x}^{2}}=\frac{y^{2}}{\sigma_{y}^{2}+{\tilde{\sigma }}_{x}^{2}\cdot y^{2}/{\hat{\mu }}_{x}^{2}}<\frac{y^{2}}{\sigma_{y}^{2}}=T.$$
That is, ${\tilde{T}}_{\mathrm{SMR}}$ is less powerful than *T*; *p*-values for both statistics are calculated from the same distribution, which is chi-square with 1 df. The larger test statistic corresponds to the smaller *p*-value.

### 2.3. Support of Profile Likelihood

We now turn to an interval estimate for $b=\mu_{y}/\mu_{x}$. Such an estimate is not trivial since the asymptotic theory is irrelevant as there is effectively only one observation in $L(\mu_{x},\;u_{y})$. For this reason, the distribution of $\hat{b}=y/{\hat{\mu }}_{x}$, which is the MLE of $b=\mu_{y}/\mu_{x}$, is far from normal. To demonstrate this point, the following simulation study is conducted.

We generate 100,000 *x*’s from a normal distribution with $\mu_{x}=4$ and $\sigma_{x}^{2}=1$ (so that there are a reasonable amount of *x*’s to be selected), 7.412 of them satisfy |x|>5.45131 and are selected. The same number (i.e., 7.412) of *y*’s are generated independently from a normal distribution with mean $\mu_{y}=b\mu_{x}$ and variance $\sigma_{y}^{2}=1$. For each (x,y) pair, $\hat{b}=y/{\hat{\mu }}_{x}$ is calculated. Histograms of $\hat{b}$ for b=0 and b=2 are shown in Figure 2. For b=0, the mean of $\hat{b}$ is 0.0073 and the median is 0.0022. The distribution has a high probability in the neighborhood of 0. For b=2, the distribution of $\hat{b}$ seems to be bimodal and is skewed to the right with a mean equal to 7.4755 and a median equal to 2.5700. Both values are larger than the true value b=2. These means and medians are also shown in Table 1 together with results from another simulation study described later.

We consider the profile log-likelihood function pl(b) defined by:$$\begin{array}{cc} pl(b)\; & =\max_{\mu_{x}}\left[\log L_{x}\left(\mu_{x}\right)+\log L_{y}\left(b\mu_{x}\right)\right]. \end{array}$$
This function is maximized at $\hat{b}=y/{\hat{\mu }}_{x}$ and the maximum is equal to $pl\left(\hat{b}\right)=\log L\left({\hat{\mu }}_{x},y\right)=\log L_{x}\left({\hat{\mu }}_{x}\right)+\log L_{y}\left(y\right)$, which is also the maximum of $\log L(\mu_{x},\mu_{y})$.

A natural interval estimate would be a 1−α profile confidence interval defined as the set of $b_{0}$ such that $H_{0}:b=b_{0}$ is not rejected at significance level α. However, the distribution of the log profile likelihood ratio
$$\begin{array}{cc} 2\left[pl\left(\hat{b}\right)-pl\left(b_{0}\right)\right]\; & =2\left[\log L\left({\hat{\mu }}_{x},y\right)-pl\left(b_{0}\right)\right] \end{array}$$
is unknown for an arbitrary $b_{0}$. The only exception is $b_{0}=0$ at which
$$\begin{array}{cc} 2\left[\log L\left({\hat{\mu }}_{x},y\right)-pl\left(0\right)\right]\; & =2\left\{\log\left[L_{x}\left({\hat{\mu }}_{x}\right)L_{y}\left(y\right)\right]-\log\left[L_{x}\left({\hat{\mu }}_{x}\right)L_{y}\left(0\right)\right]\right\}\\ \; & =2\left[\log L_{y}\left(y\right)-\log L_{y}\left(0\right)\right]\\ \; & =\frac{y^{2}}{\sigma_{y}^{2}}∼\chi_{1}^{2}. \end{array}$$
Intuitively, the log partial likelihood function pl(b) can not be approximated by a quadratic function in the vicinity of $\hat{b}$ when $b_{0}\neq 0$. As a result, the profile confidence interval for *b* can not be constructed. An example log partial likelihood function pl(b) is shown in Figure 3 for $x/\sigma_{x}=5.4599$ and $y/\sigma_{y}=12.3155$.

For an interval estimate of *b*, we use the *k*-unit support defined by [27]:$$\left\{b_{0}:pl\left(\hat{b}\right)-pl\left(b_{0}\right)<k\right\}=\left\{b_{0}:pl\left(b_{0}\right)>\log L\left({\hat{\mu }}_{x},y\right)-k\right\},$$
where *k* is a prespecified number. This interval consists of $b_{0}$ for which $pl(b_{0})$ is greater than $\log L({\hat{\mu }}_{x},y)-k$. It can be regarded as a generalization of the usual confidence interval. For instance, when $b_{0}=0$ and k=2,
$$\begin{array}{cc} 0.95 & =\mathrm{Pr}(2\left[\log L\left({\hat{\mu }}_{x},y\right)-pl\left(0\right)\right]<3.84)\\ & =\mathrm{Pr}(\log L\left({\hat{\mu }}_{x},y\right)-pl\left(0\right)<1.92)\\ \; & \approx \mathrm{Pr}(\log L\left({\hat{\mu }}_{x},y\right)-pl\left(0\right)<2). \end{array}$$
This approximation worsens as $|b_{0}|$ moves further away from 0. For the example shown in Figure 3 (i.e., $x/\sigma_{x}=5.4599$ and $y/\sigma_{y}=12.3155$), the lower limit of the 2-unit support is 2.146 and the upper limit is greater than 43.406. The exact value of the upper limit is unknown due to numerical issues. It may be unbounded.

By the way the support is constructed, the null $H_{0}:b_{0}=0$ is rejected by the statistic *T* at significance level α if and only if the *k*-unit support, where $k={\left[\Phi^{-1}\left(1-\alpha /2\right)\right]}^{2}/2$, contains 0.

We use a simulation study to investigate the coverage of a 2-unit support. For this purpose, data are generated as before but more values for *b*, i.e., b=0, 0.5, 1, 1.5, and 2, are considered. For each value of *b*, we compare the winner’s-0curse-corrected method and the SMR method in terms of a point estimate of *b*, an interval estimate of *b*, and a test of $H_{0}:b=0$. Results are reported in Table 1. Both the winner’s-curse-corrected method and SMR method are biased in terms of the mean and median. The SMR 95% confidence interval, computed as ${\hat{b}}_{\mathrm{SMR}}\pm 1.96\times V_{\mathrm{delta}}^{1/2}$, has worse coverage as *b* increases while the 2-unit support has rather stable coverage. In addition, the test statistic *T* is more powerful than the SMR method.

## 3. An Empirical Data Analysis

We conducted a Mendelian randomization analysis of the effect of age of menarche on total pubertal height growth and late pubertal height growth using the winner’s-curse-corrected method and the SMR method. Previously, we used the inverse-variance weighted (IVW) method [5] and the MR-Egger regression method [6] on these exposures and outcomes [15]. In that study, to avoid the winner’s curse caused by the selection of IV SNPs, two other GWAS studies on age at menarche from an MR-Base database were used for validation. IV SNPs were significant in the main GWAS for age at menarche but not in the other two other GWASs which were removed. Such a procedure helps to avoid IV SNPs that are close to the selection threshold. In this study, we use all significant IV SNPs without further validation.

GWAS summary data were retrieved from the MR-Base database (http://www.mrbase.org/ accessed on 27 November 2022). At the genome-wide significance level $5\times 10^{-8}$, 117 instrument SNPs were selected from a previous study on age at menarche with 182,413 females of European ancestry [28]. After pruning for linkage disequilibrium, there are 84 SNPs left. The GWAS summary statistics on adult height were obtained from a study with 4946 females of European ancestry [29]. Thus, the population of this study matches that of the study on age at menarche.

For each SNP, the winner’s-curse-corrected estimate of *b* and a support are computed in addition to the SMR estimate and the associated confidence interval. To correct for the 84 IV SNPs, the support is 5.9-unit since $\mathrm{Pr}(\chi_{1}^{2}>2\times 5.9)=0.05/84$ and the nominal coverage of the confidence interval is 0.9994(=1−0.05/84). As discussed previously, this support excludes 0 if and only if the *T* statistic is significant at the level 0.05/84. The *p*-value for the winner’s-curse-corrected method is based on the *T* statistic. SNPs whose supports or confidence intervals do not contain 0 are shown in Table 2.

The estimates of *b* from the winner’s-curse-corrected method and the SMR method are pretty close to each other for the SNPs shown in Table 2, as are the support and the confidence interval. This is due to the high significance of the association of these SNPs with the age at menarche (*p*-values: $4.552\times 10^{-15}$ for rs7514705 and rs7642134; $<4.552\times 10^{-15}$ for rs7759938). For both total and late pubertal height growth, the *T* statistic is more significant than the $T_{\mathrm{SMR}}$ statistic. For late pubertal height growth, SNP rs7514705 is significant for the *T* statistic but not for the $T_{\mathrm{SMR}}$ statistic.

Another empirical application on the conditional test *T* is the study of schizophrenia, which was shown in our previous publication [9]. The *T* statistic identified some strong candidate genes (e.g., AKT3, RGS6, and KCNN3) for schizophrenia that are missed by the SMR method.

## 4. Discussion

Previously, we proposed a test statistic *T* for testing $H_{0}:b=0$ [15]. The current work extends the previous work by focusing on the point and interval estimate of the causal effect *b*. Because the “sample size” for the MR analysis is 1, the standard likelihood theory does not apply. As a result, it is not straightforward to construct a confidence interval.

We considered two extreme scenarios: one being the one-sample individual-level data and the other being independent-sample summary data. In addition, on of these scenarios is without the winner’s curse caused by the selection of IV SNPs and the other suffers from the winner’s curse. For one-sample individual-level data that is free of the winner’s curse, the SMR method is the same as the TSLS method, not only in terms of the estimates of the causal effect size but also in terms of the variance of the estimates. For two independent-sample summary data with a selected SNP, the SMR test for $H_{0}:b=0$ is less powerful than the conditional test we proposed earlier [9]. Confidence intervals derived from the SMR method have poor coverage compared to their nominal levels. In comparison, the supports we proposed have stable coverage, at least in our simulation studies.

There are reports (also see our empirical data analyses) showing that the winner’s curse may not have substantial impact on the MR estimates [12]. This is because in these cases the SNPs are strong. As indicated by Figure 1, the winner’s curse affects the relatively weak IV SNPs most. These are the SNPs with $|{\hat{b}}_{gx}/SE\left({\hat{b}}_{gx}\right)|<7.5$ (i.e., $p<6.38\times 10^{-14}$). For the strong SNPs, there is not much difference between *x* and its maximum likelihood estimate. An SNP is strong when either the effect size *b* or the sample size in the exposure GWAS is, or both, are large. The three-sample design [14] also helps in making an SNP strong by increasing the chance that the selected SNPs are highly significant. Theoretically, however, it does not eliminate the winner’s curse as the probability that a weak SNP is significant in the discovery GWAS *and* the exposure GWAS is non-zero.

In the previous paragraph, the meaning of the term “weak” may be different than weak instrument in the usual sense although there is no universally-accepted definition of weak instrument. It is relative to the threshold for selecting IV SNPs. SNPs that barely pass the threshold are always weak. In comparison, a weak instrument in the usual sense seems to be characterized in absolute sense, for instance, the *F*-statistic for testing $H_{0}:b_{gx}=0$ is less than 10 [30].

Although Equation (Equation 1) is on continuous traits, the proposed winner’s-curse-corrected method works for dichotomous traits because it is based on the approximate normality on ${\hat{b}}_{gx}$ and ${\hat{b}}_{gy}$.

The current study focuses on a single SNP analysis. A major advantage of such an analysis over multiple SNPs such as the IVW method and the MR-Egger regression method is that it involves less assumptions. For example, the causal effects at different SNPs are allowed to be different. An interesting topic would be to generalize the current work to the case of using multiple SNPs simultaneously.

Our winner’s-curse-corrected method is designed for two independent (i.e., non-overlapping) samples only. This is a limitation although it is not uncommon for methodology development, for instance, [14]. In practice, the study subjects for the exposure GWAS and the outcome GWAS may overlap [12]. The likelihood function $L(\mu_{x},\mu_{y})$ will be different than what is presented here. The conditional test *T* needs to be revised and the concept of support is still applicable. Future research on this topic is warranted.

The winner’s-curse-corrected method has been implemented in the R package iGasso.

## Figures and Tables

**Figure 1 genes-14-00211-f001:**
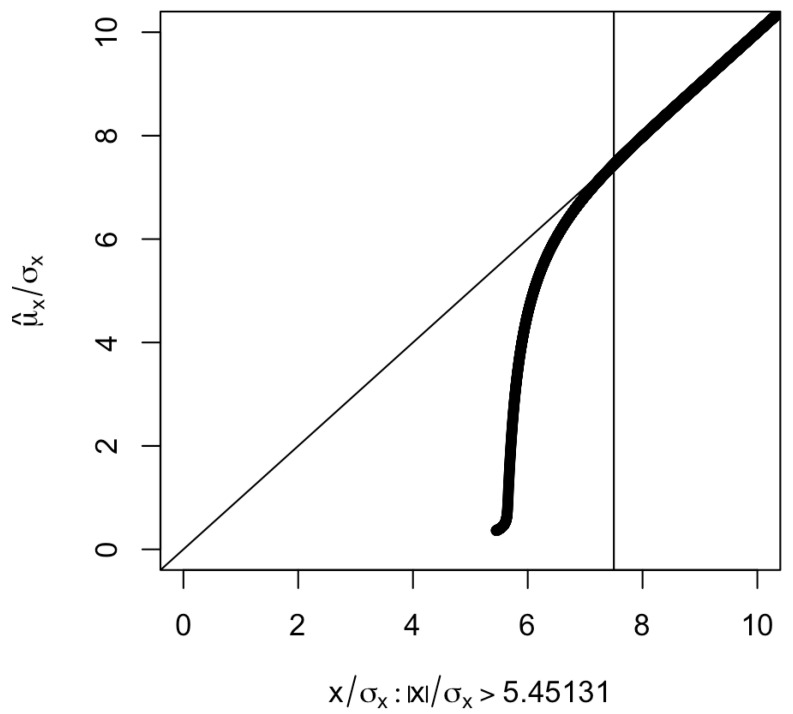
Plot of ${\hat{\mu }}_{x}/\sigma_{x}$ against $x/\sigma_{x}$ selected under $|x/\sigma_{x}|>5.45131$ (corresponding to $p<5\times 10^{-8}$). The vertical line is at $|x/\sigma_{x}|=7.5$, which corresponds to a *p*-value of $6.38\times 10^{-14}$. The part corresponding to $x/\sigma_{x}<0$ is not shown since ${\hat{\mu }}_{x}/\sigma_{x}$ is an odd function of $x/\sigma_{x}$.

**Figure 2 genes-14-00211-f002:**
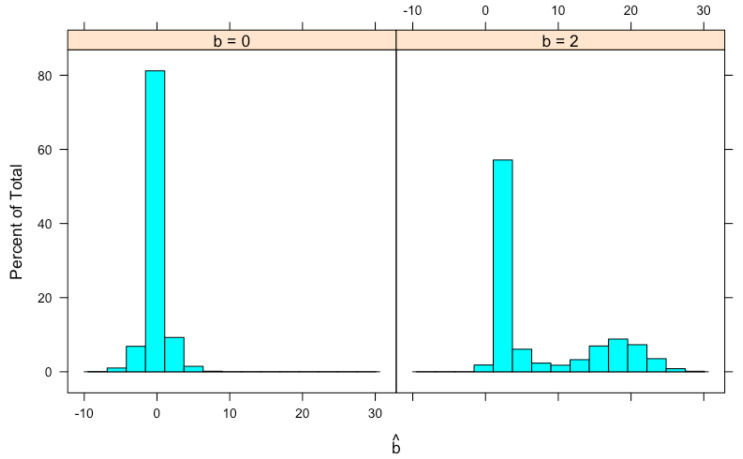
Histogram of simulated $\hat{b}=y/{\hat{\mu }}_{x}$, the MLE of *b*. Data simulation procedure is described in the text.

**Figure 3 genes-14-00211-f003:**
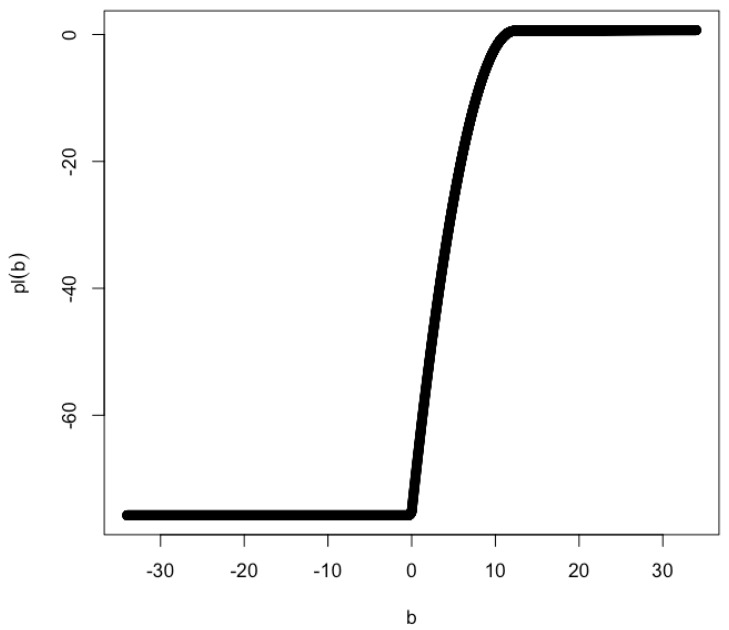
Profile likelihood for $x/\sigma_{x}=5.4599$ and $y/\sigma_{y}=12.3155$. The MLE of *b* is $\hat{b}=33.416$. The lower limit of the 2-unit support is 2.146 and the upper limit is greater than 43.406. The exact value of the upper limit is unknown due to numerical issues. It may be unbounded.

**Table 1 genes-14-00211-t001:** Results of simulation studies with $\mu_{x}=4$ and $\sigma_{x}=\sigma_{y}=1$. The statistic *T* is $T=y^{2}/\sigma_{y}^{2}$.

	*b*				
Method	0	0.5	1	1.5	2
Winner’s-curse-corrected					
Mean of $\hat{b}$	0.0073	1.8743	3.7414	5.6084	7.4755
Median of $\hat{b}$	0.0022	0.6843	1.3091	1.9327	2.5700
Coverage of 2-unit support	0.9587	0.9725	0.9803	0.9816	0.9811
Power of *T* for testing $H_{0}:b=0$	0.0471	0.5217	0.9807	1.0000	1.0000
SMR					
Mean of ${\hat{b}}_{\mathrm{SMR}}$	0.0019	0.3424	0.6829	1.0234	1.3639
Median of ${\hat{b}}_{\mathrm{SMR}}$	−0.3310	0.3405	0.6795	1.0199	1.3615
Coverage of 95% CI	0.9648	0.8524	0.6511	0.4966	0.3958
Power for testing $H_{0}:b=0$	0.0353	0.4721	0.9726	1.0000	1.0000

**Table 2 genes-14-00211-t002:** Results for the effects of age at menarche on total pubertal height growth and late pubertal height growth. To correct for the 84 IV SNPs, the support is 5.9-unit and the nominal coverage of the CI is 0.9994(=1−0.05/84). This support excludes 0 if and only if the *T* statistic is significant at the level 0.05/84. The *p*-value is for the null $H_{0}:b=0$. It is computed from the *T* statistic (the winner’s-curse-corrected method) or the $T_{\mathrm{SMR}}$ statistic (the SMR method).

Winner’s-Curse-Corrected Method			
**SNP**	**Gene Name**	**$\hat{b}$ (5.9-Unit Support)**	* **p** * **-Value**
**Total pubertal height growth**			
rs7514705	TNNI3K	2.048 (0.889, 3.807)	$8.856\times 10^{-6}$
rs7642134	POU1F1	2.474 (1.264, 4.433)	$1.117\times 10^{-7}$
**Late pubertal height growth**			
rs7514705	TNNI3K	1.822 (0.057, 5.091)	$5.024\times 10^{-4}$
rs7759938	LIN28B	0.931 (0.335, 1.571)	$2.756\times 10^{-7}$
**SMR Method**			
**SNP**	**Gene Name**	**${\hat{b}}_{\mathrm{SMR}}$ (99.94% CI)**	* **p** * **-Value**
**Total pubertal height growth**			
rs7514705	TNNI3K	2.042 (0.330, 3.754)	$1.108\times 10^{-4}$
rs7642134	POU1F1	2.466 (0.647, 4.284)	$1.110\times 10^{-5}$
**Late pubertal height growth**			
rs7759938	LIN28B	0.931 (0.330, 1.533)	$5.142\times 10^{-7}$

## Data Availability

GWAS summary data used in this research were retrieved from the MR-Base database (http://www.mrbase.org/ accessed on 27 November 2022).

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
