# Peer review of "Support Interval for Two-Sample Summary Data-Based Mendelian Randomization"

_genes, 2023, doi:10.3390/genes14010211_

Round 1
Reviewer 1 Report
This manuscript is generally well-written and well-structured. In this manuscript, the author proposed a novel selection-corrected two-sample summary-data-based Mendelian randomization method. This method could help address the selection bias that occurred in instrumental SNP selection based on GWAS results.
Major Comments:
1. Introduction section seems to be a bit brief. For the general audience, they may need more background knowledge about MR.
2. For the comparison between Tsmr and T from line 99 to line 100, can you elaborate more on why Tsmr is less powerful than T.
3. Line 107, for simulation study, it might be better to replace one hundred thousand with 100,000, same for seven thousand four hundred and twelve.
4. Line 107, could you explain the reasons for these parameter selections? Do they match the estimates from GWAS analysis?
5. For the general audience, more introduction of the method of support in the introduction part will be helpful.
6. For methods to be generally more powerful than SMR, more studies should be conducted to show the overall performance. Have you tried other traits?
Author Response
We sincerely thank you for your time evaluating our work. Our responses are given below. For your convenience, your comments are italicized and included as well.
- Introduction section seems to be a bit brief. For the general audience, they may need more background knowledge about MR.
Thank you for your suggestion. More background information on MR is added (lines 23-25, lines 36-41).
- For the comparison between Tsmrand T from line 99 to line 100, can you elaborate more on why Tsmr is less powerful than T.
More explanation is added (lines 113-115). Basically, Tsmr and T use the same distribution for p-value. The larger the value the more powerful the test statistic is and T is larger than Tsmr.
- Line 107, for simulation study, it might be better to replace one hundred thousand with 100,000, same for seven thousand four hundred and twelve.
These sentences have been restructured so that these numbers are no longer at the beginning of these sentences. In the restructured sentences, these numbers are expressed in numerals as you suggested.
- Line 107, could you explain the reasons for these parameter selections? Do they match the estimates from GWAS analysis?
These parameter values are inspired by GWAS analysis (the true values underlying a GWAS are unknown by nature). They are chosen so that the probability of being selected is not low. A sentence is added to explain this choice (lines 122-123). Personally, I don’t think the values used matter much for our purpose.
- For the general audience, more introduction of the method of support in the introduction part will be helpful.
Two sentences have been added to explain the method of support (lines 54-56).
- For methods to be generally more powerful than SMR, more studies should be conducted to show the overall performance. Have you tried other traits?
In Wang and Han (2021) we studied schizophrenia using the conditional test. This study is mentioned in lines 183-184.
Reviewer 2 Report
Thank you for inviting me to review this interesting article " Correcting selection bias in two sample summary data based Mendelian randomization". The authors aim to introduce a new statistical correction for Winner's curse. This is an important issue in MR and this is a novel and interesting approach to address it. The authors propose a novel sensitivity analysis which uses maximum likelihood estimation to try to correct for bias in classical wald ratios.
My main issue this the article is the misleading title: in the MR literature, and statistical genetics/epidemiology more broadly, selection bias typically refers to bias introduced by the selection of participants, whereas bias due to the selection of genetic variants is called Winner's curse. The latter term is not used in the article, and you do not define what you mean by selection bias. I would suggest that you clarify this in the introduction and at a minimum say that what you mean by selection bias is typically called Winner’s Curse.
The above confusion has resulted in the authors missing important parts of the literature. Firstly, 'three-sample' MR solves the issues with Winner's curse, and there are already existing corrections (such as https://academic.oup.com/bioinformatics/article/32/17/2598/2450747?login=false) which you do not discuss. Likewise, there is decent evidence that when selecting genome wide sig. SNPs the bias is quite small anyway (https://www.medrxiv.org/content/10.1101/2022.08.05.22278470v1). Which, is what you find in your applied example. These should be incorporated into the discussion.
MR studies often have overlapping samples. But, in the derivation of the sensitivity analysis you say that you assume no sample overlap so that the covariance can be assumed to be zero. I think it would be worth adding either a simulation or a formal derivation of how the test would be biased with sample overlap.
On line 78 you say that the mean of x and the mean of y are independent. This only works if the null hypothesis is true. Could this account for the large inflation in the mean of b^ you observe in your simulation? and if so, how will you address it?
Finally, on a practical note, I think the manuscript is way to technical for the average MR user. I would suggest (e.g. in the discussion) that the authors add a lay summary of how the method works and the assumptions it makes and its limitations. Likewise, have the authors considered creating an R package to help facilitate its implementation.
Author Response
We sincerely appreciate your feedback. Substantial changes have been made to address your comments. Our point-to-point responses are given below. For your convenience, your comments are italicized and included as well.
My main issue this the article is the misleading title: in the MR literature, and statistical genetics/epidemiology more broadly, selection bias typically refers to bias introduced by the selection of participants, whereas bias due to the selection of genetic variants is called Winner's curse. The latter term is not used in the article, and you do not define what you mean by selection bias. I would suggest that you clarify this in the introduction and at a minimum say that what you mean by selection bias is typically called Winner’s Curse.
Thank you for your point. This term has been used throughout.
The above confusion has resulted in the authors missing important parts of the literature. Firstly, 'three-sample' MR solves the issues with Winner's curse, and there are already existing corrections (such as https://academic.oup.com/bioinformatics/article/32/17/2598/2450747?login=false) which you do not discuss. Likewise, there is decent evidence that when selecting genome wide sig. SNPs the bias is quite small anyway (https://www.medrxiv.org/content/10.1101/2022.08.05.22278470v1). Which, is what you find in your applied example. These should be incorporated into the discussion.
Thank you for bringing my attention to this literature. The Discussion section is almost completely rewritten to incorporate the publications you mentioned. There is a lengthy discussion on the bias caused by the winner’s curse in light of the results in the current work (lines 201-215).
MR studies often have overlapping samples. But, in the derivation of the sensitivity analysis you say that you assume no sample overlap so that the covariance can be assumed to be zero. I think it would be worth adding either a simulation or a formal derivation of how the test would be biased with sample overlap.
You made an important point. A discussion on this issue is added to the Discussion section (lines 224-229). Considering the complexity of possible overlapping patterns (https://www.medrxiv.org/content/10.1101/2022.08.05.22278470v1), a formal treatment is better left to future research. I hope the discussion (i.e., lines 224-229) answers your questions well. If not, please let me know.
On line 78 you say that the mean of x and the mean of y are independent. This only works if the null hypothesis is true. Could this account for the large inflation in the mean of b^ you observe in your simulation? and if so, how will you address it?
The large inflation in the mean of is caused by weak SNPs that barely pass the selection threshold. For such SNPs, the effect of winner’s curse is strong (Figure 1), which leads to large values of . For strong IV SNPs, the effect of winner’s curse is ignorable (Figure 1). On average, the mean of is high. If the exposure GWAS does not overlap with the discovery GWAS, winner’s curse will disappear. Also see the discussion on lines 224-229.
Finally, on a practical note, I think the manuscript is way to technical for the average MR user. I would suggest (e.g. in the discussion) that the authors add a lay summary of how the method works and the assumptions it makes and its limitations. Likewise, have the authors considered creating an R package to help facilitate its implementation.
A summary is provided in the Discussion section (lines 191-200). The proposed method has been implemented in an R package named iGasso, which is mentioned at the very end of the Discussion section.
Round 2
Reviewer 1 Report
Thanks for answering the questions.
Considering the audience for the journal, did any of your analyses find that the method can help identify new associations? If that's the case, it would be more convincing.
Author Response
Thank for your point. In the real data analysis, we emphasized that for late pubertal height growth, gene TNN13K is identified by our T test but not by the SMR method (lines 182-184). We also emphasized that in the study of schizophrenia presented in our previous publication [9], three genes are identified by our T test but not by the SMR method (lines 186-188).
Happy New Year,
Kai
